# Diagnostic and Therapeutic Challenge Caused by *Candida albicans* and *Aspergillus* spp. Infections in a Pediatric Patient as a Complication of Acute Lymphoblastic Leukemia Treatment: A Case Report and Literature Review

**DOI:** 10.3390/pathogens13090772

**Published:** 2024-09-07

**Authors:** Natalia Zaj, Weronika Kopyt, Emilia Kamizela, Julia Zarychta, Adrian Kowalczyk, Monika Lejman, Joanna Zawitkowska

**Affiliations:** 1Student Scientific Society of Department of Pediatric Hematology, Oncology and Transplantology, Medical University of Lublin, 20-093 Lublin, Poland; natia.zaj@gmail.com (N.Z.); weronikakopyt1@gmail.com (W.K.); kamizela2000@gmail.com (E.K.); julia.zarychta99@gmail.com (J.Z.); adriankowalczyk31@gmail.com (A.K.); 2Independent Laboratory of Genetic Diagnostics, Medical University of Lublin, 20-093 Lublin, Poland; monika.lejman@umlub.pl; 3Department of Pediatric Hematology, Oncology and Transplantology, Medical University of Lublin, 20-093 Lublin, Poland

**Keywords:** invasive fungal infections, *Aspergillus* spp. infection, *Candida albicans* infection, children, acute lymphoblastic leukemia

## Abstract

Fungal infections constitute a significant challenge and continue to be a predominant cause of treatment failure in pediatric leukemia cases. Despite the implementation of antifungal prophylaxis, these infections contribute to approximately 20% of cases in children undergoing treatment for acute lymphoblastic leukemia (ALL). The aim of this study is to highlight the diagnostic and therapeutic challenges associated with invasive fungal infections (IFIs). We also present a review of the epidemiology, risk factors, treatment, and a clinical presentation of IFI in patients with ALL. This case report details the clinical course of confirmed *Candida albicans* (*C. albicans*) and *Aspergillus* spp. infections during the consolidation phase of ALL treatment in a 5-year-old pediatric patient. This male patient did not experience any complications until Day 28 of protocol II. Then, the patient’s condition deteriorated. Blood culture detected the growth of *C. albicans*. Despite the implementation of targeted therapy, the boy’s condition did not show improvement. The appearance of respiratory symptoms necessitated a computed tomography (CT) of the chest, which revealed multiple nodular densities atypical for *C. albicans* etiology. In spite of ongoing antifungal treatment, the lesions depicted in the CT scans showed no regression. A lung biopsy ultimately identified *Aspergillus* species as the source of the infection. Overcoming fungal infections poses a considerable challenge; therefore, an accurate diagnosis and the prompt initiation of targeted therapy are crucial in managing these infections in patients with leukemia.

## 1. Introduction

The tremendous progress made in medicine has transformed acute lymphoblastic leukemia (ALL) from a typically fatal disease into a curable one. Currently, the survival rate among pediatric patients is around 90% [1,2]. This is possible because of current global treatment programs that take into account the assessment of the risk of leukemia relapse based on the clinical picture, response to chemotherapy, and molecular defects [3,4].

Despite significant progress in treatment and increased survival rates, complications related to chemotherapy have become a challenge [5]. An analysis of the Polish documentation from 2012 to 2017, collected on a biennial basis, showed that 53.2% (726/1363) children with newly diagnosed ALL experienced a bacterial infection. The total number of episodes amounted to 1511. Viral and fungal infections affected 18.4% (251/1363) and 20.4% (278/1363) of pediatric patients, respectively [6]. In the next study from 2020 to 2021, 78.5% (361/460) of children were reported as having a bacterial infection during the therapy of ALL. The percentage of fungal infections was similar to the previous study and this time reached a high rate of 21.5% (99/460) [7]. Viral infections also represented a large group of infectious complications (37.5%; 192/510), and in 2020–2021, the most common viral illness was coronavirus disease 2019 (COVID-19) [8]. According to literature findings, mortality related to infections (IRMs) among patients with ALL constitutes 30% of all deaths and 64% of treatment-related mortality. Bacterial infections are the leading cause of mortality (68%), followed by fungal (20%) and viral infections (12%) [7,9]. Lehrnbecher et al., in an analysis of the international, multi-center prospective randomized Phase III clinical trial AIEOP-BFM ALL 2009, showed that mortality from the invasive fungal infections (IFI) at 6 weeks and 12 weeks was 10.7% and 11.2%, respectively [10]. An analysis of the data from the General University Hospital Gregorio Marañón in Madrid, Spain, reports mortality from IFI at a rate of 21.4% [11]. In another analysis, the mortality rate was 38.5% [12]. This indicates that IFIs in patients with hematologic diseases are common and associated with high mortality. Overcoming fungal infections poses a considerable challenge. An accurate diagnosis and the prompt initiation of targeted therapy are crucial in managing these infections in patients with leukemia.

In this work, we present the case report of a 5-year-old boy who suffers from ALL of B-cell precursor origin. The patient had *Candida albicans* (*C. albicans*) and *Aspergillus* spp. infections confirmed during the consolidation phase of ALL treatment. The aim of this study is to highlight the diagnostic and therapeutic challenges associated with IFI. We also present a review of the epidemiology and risk factors for IFI in patients with ALL undergoing chemotherapy, as well as the clinical presentation of the infection and the applied treatment. 

## 2. Materials and Methods

A PubMed, Google Scholar and Scopus search for was performed using the terms “*Aspergillus* spp. infection”, “*C. Albicans* infection”, “children”, “acute lymphoblastic leukemia”, and “invasive fungal disease”. A literature review was created based on available published data, which was released between 2000 and 2024. Only full-text articles in English were included. Articles in other languages, not applicable to the pediatric population, articles that were primarily theoretical or non-clinical were excluded from the review. The review included case reports, literature reviews and meta-analyses.

## 3. Case Presentation

A 5-year-old patient was diagnosed with precursor B-cell ALL (pB-ALL) on 6 March 2023. The patient was treated according to the treatment protocol for children and adolescents with pB-ALL currently applicable in Poland. The therapy schedule for this child included the following: an induction of remission (protocol I: steroids, vincristine, daunorubicin (four doses), and PEG L-asparaginase (two doses); consolidation: arabinoside cytosine, cyclophosphamide, and mercaptopurine), protocol M (methotrexate 5 g/m^2^ infusion (four doses), intrathecal methotrexate (four doses), and mercaptopurin), protocol II (dexamethasone, vincristine (four doses), doxorubicine (four doses), PEG L-asparaginase (one dose), arabinoside cytosine, cyclophosphamide, and thioguanine), and maintenance therapy (oral mercaptopurin every day and methotrexate one day per week) [13]. The child was assigned to a standard risk group and received fluconazole as antifungal prophylaxis and oral cotrimoxazole as prophylaxis against *Pneumocystis jirovecii* (*P. jirovecii*) infection. The induction phase, protocol M and protocol II (up to Day 28) continued without serious complications [13]. 

### 3.1. The First Episode of Infection

On Day 29 of protocol II, a sudden deterioration in the patient’s condition was observed. The child experienced weakness and a fever that reached 38.5 degrees Celsius. The laboratory test results revealed a neutropenia level of 0.09 × 10^3^/µL (3.4–9.5 × 10^3^/µL reference range), hypertriglyceridemia (412 mg/dL; normal: <150 mg/mL), and thrombocytopenia 30 × 10^3^/µL (140–410 × 10^3^/µL reference range). Moreover, an elevated procalcitonin level reaching 0.89 ng/mL (normal: <0.5 ng/mL) and a C-reactive protein (CRP) level of 17.67 mg/dL (0–0.5 mg/dL reference range) were detected. This led to the decision to discontinue chemotherapy administration [13]. Empiric therapy initially included liposomal amphotericin B [14], meropenem, and vancomycin [15,16]. Blood culture taken from the central venous catheter (CVC) showed a growth of *C. albicans*. Due to these results, the CVC was urgently removed. Simultaneously an Arrow type of catheter was inserted into the femoral vein. Supportive care included the transfusion of blood products (platelet concentrate and red blood cells), intravenous immunoglobulin administration (0.4 g/kg/d), and subcutaneous granulocyte growth factor administration (5 µg/kg/d). After four days from the onset of infection symptoms, an increased respiratory effort was observed. An extended microbiological diagnostics for lower respiratory tract infections excluded *Chlamydia pneumoniae*, *Legionella pneumophila*, and *Mycoplasma pneumoniae* as potential infectious agents. Elevated levels of *Pneumocystis* IgM antibodies were detected. Subsequently, therapeutic doses of trimethoprim–sulfamethoxazole were used in the treatment against *P. jirovecii*. Chest computed tomography (CT) was performed, but the result did not reveal any abnormalities. On Day 8 of persistent fever, a gradual growth in the CRP level was observed. The boy’s general condition was still average; he was consistently experiencing weakness as well as prolonged fever. On Day 24 from the first symptoms, a follow-up blood culture was performed and *C. albicans* was constantly detected. The central catheter was removed for the second time due to the suspicion of another infection. A peripheral catheter was inserted. Despite the administered treatment, the boy’s condition continued to deteriorate. Given the respiratory failure, the child required passive oxygen therapy at a flow of three liters per minute. An auscultation revealed that the patient had crackling sounds over the right lung. A dry cough was also observed during the examination. The signs of respiratory failure led to a decision to carry out a further CT examination. The result of the chest tomography, performed only 11 days later vs. the previous one, turned out to be abnormal. The radiological report matched scattered nodular and small nodular densities, which were bilaterally distributed and uncountable. The largest subpleural lesions showed a tendency towards greater consolidation, surrounded by ground-glass opacification (a halo sign visible). An intensification of lesions was especially located in the lower lobes (Figure 1A). 

These findings resulted in the discontinuation of liposomal amphotericin B administration after 13 days of use and the implementation of caspofungin. On Day 18 of the occurrence of infection symptoms, fever and elevated CRP levels were still present, which led to the decision to discontinue vancomycin and meropenem administration. Tigecycline, colistin, and amikacin were used instead in the treatment. The aforementioned treatment modifications are presented in Figure 2. 

On Day 26, the patient’s condition improved. The laboratory test results showed that the CRP level was decreasing. The persistent fever was a disturbing symptom. Due to the suspicion of an inflammatory etiology, dexamethasone was introduced to the treatment. The third CT of the chest was performed. The lesions resembling the ones from the previous CT results were present, but this time, the image demonstrated a regression of the lesions. The implementation of caspofungin in the treatment as well as the initiation of steroid treatment were crucial in overcoming the first infection episode.

### 3.2. Transitional Period between Infections

The fever and signs of respiratory failure disappeared. Antibiotics were discontinued, while caspofungin administration was still maintained. The decision to resume the administration of cytostatics, on Day 36 of protocol II, was made. After a two weeks, dexamethasone administration was stopped.

### 3.3. The Second Episode of Infection

Two days after dexamethasone discontinuation, the symptoms of infection returned. Fever, elevated inflammatory markers and episodes of respiratory failure were observed again. Chemotherapy administration was stopped on Day 40 of protocol II. An empirical antibiotic therapy was reintroduced. Piperacillin with tazobactam and vancomycin were used at the time. Voriconazole was added to treatment as an antifungal agent. A CT examination was requested and revealed the progression of lesions compared to the previous CT examination. Simultaneously, the parainfluenza virus was detected in a respiratory panel performed for upper respiratory tract infections. A blood culture performed the next day showed the presence of *Morganella morgani*, *Enterococcus faecalis*, and *Staphylococcus haemolyticus* that was MRCNS-positive (methicillin-resistant coagulase-negative *Staphylococcus*). After six days of the second episode duration, due to the persistent symptoms of the infection, blood and stool cultures were performed again. The growth of MRCNS-positive *Staphylococcus hominis* was observed in the blood culture. The growth of *Klebsiella pneumoniae* ESBL (extended-spectrum beta-lactamase)-positive as well as ESBL-positive *Citobacter freundii* and *Enterococcus* spp. were present in the stool culture. The targeted therapy is presented in Figure 3. 

The patient’s general condition improved with the normalization of inflammatory markers parameters. On Day 14 of the second episode of infection, the decision was made to discontinue antibiotic administration, but on the next day, the symptoms returned. Antibiotics were reintroduced into the treatment. Tigecycline, amikacin, and rifampicin were implemented. The choice of tigecycline and amikacin was based on a satisfactory response to these antibiotics’ implementation during the first infectious episode. Rifampicin was administered as an antibiotic with high penetration into tissues and organs [17]. The implemented therapy was effective against both Gram-positive and Gram-negative bacterial groups [18,19,20]. The follow-up CT examination demonstrated the progression of lesions. The largest lesions were present at the base of the lower lobes with a tendency towards consolidation. In order to control the primary disease, a bone marrow biopsy was conducted, and the results confirmed remission. Despite the implementation of broad-spectrum antifungal and antibiotic therapy, symptoms such as a fever, elevated inflammatory markers, and respiratory failure were consistently present. Numerous tests were performed to search for the source of infection in the second episode, as well as in the previous one. Hemophagocytic lymphohistiocytosis (HLH), macrophage activation syndrome (MAS) and tuberculosis were excluded in the differential diagnosis. Differential diagnosis details and the conducted tests are presented in Figure 4. 

Despite the use of antibiotic therapy based on the antibiogram results, the treatment still did not bring the expected long-term results. It was decided to perform bronchoscopy and lung biopsy. In the Department of Thoracic Surgery, Video-Assisted Thoracoscopic Lung Biopsy was performed under general anesthesia. During the procedure, the lingula of the lung with visible nodules, size 4 × 2.5 × 1 cm, was resected and then examined histopathologically. The result of the examination indicated the presence of granulomatous nodules, composed of histiocytes. Necrosis and signs of abscess were visible in the center of the nodules. Numerous neutrophils and eosinophils were present within the abscesses. The periphery of the nodules was marked by lymphocytes and plasma cells, as well as fibrosis. Spores and mycelium threads similar to *Aspergillus* species were observed in the microscopic analysis as well. During hospitalization, antibiotic therapy, steroid therapy and antifungal treatment were continued. The child’s condition improved. After six days, the patient was discharged from hospital in a good general condition. Cotrimoxazole and posaconazole at a therapeutic dose, as well as dexamethasone and rifampicin, were administered orally as antimicrobial prophylaxis. Thirteen weeks from the onset of the first infectious episode, the patient was electively admitted to the clinic for a follow-up chest CT scan and immunoglobulin administration. The CT showed the partial regression of the lesions (Figure 1B) compared to the previous results. Due to severe complications, the decision was made not to continue intensive chemotherapy. Maintenance treatment was initiated at a dose of 50% 6-mercaptopurine, fourteen weeks after the onset of the first infectious episode. Currently, the boy is in a good general condition and he is receiving maintenance therapy. 

## 4. Discussion

The co-occurrence of infections caused by different etiological factors during oncological treatment is a significant challenge [21,22,23]. In the described case, after Day 28 of protocol II, the child developed two fungal infections caused by different etiological factors during the treatment of precursor B-cell ALL: *Candida albicans* and *Aspergillus* spp. According to epidemiological data, the aforementioned fungal species are the most common etiological factors of IFI. However, it is worth emphasizing that although *C. albicans* is still the main *Candida* spp. associated with the development of IFI in pediatric patients, in recent years, there has been an increase in the number of fungal infections caused by other *Candida* spp.: *Candida parapsilosis*, *Candida glabrata*, *Candida tropicalis* and *Candida krusei* [24,25,26,27]. The high prevalence of non-*C. albicans Candida* (NCAC) species may be related to higher resistance to certain antifungal drugs [27,28]. In our case, the diagnostic process did not identify the *Aspergillus* spp. responsible for the development of the infection. However, the most common strains of *Aspergillus* observed in the pediatric population are *A. fumigatus*, *A. nidulans*, *A. flavus*, *A. terreus* and *A. niger* [29]. 

Pediatric patients with proliferative diseases affecting bone marrow are particularly susceptible to the development of IFI due to reduced immunity resulting from the anticancer treatment and the underlying disease itself, which can lead to bone marrow failure [30,31,32]. A reduced neutrophil count has been associated with over 50% of IFI episodes, and therefore, recurrent or longer than 10-day episodes of neutropenia are one of the key risk factors for the development of IFI [33,34]. It is worth highlighting that over half of the pediatric patients treated for cancer have at least one episode of febrile neutropenia during therapy [35]. Another risk factor for the development of IFI, which also occurred in our patient, was the implanted CVC. Blyth et al. indicated that the use of CVC was associated with the development of candidemia in 70% of cases in children and 58% in neonates [36]. Other factors increasing the risk of developing IFI include the following: the use of steroid therapy (especially if it lasts more than 3 months), enteral nutrition, recent surgery and a bacterial infection treated with antibiotics [37,38,39,40,41]. The use of broad-spectrum bactericidal preparations, steroids and chemotherapeutics may lead to a decrease in the number of commensal bacteria and fungi and thus contribute to increased fungal colonization, which results from a decrease in the competitive pressure exerted by the physiologically occurring bacterial flora [42,43,44,45,46]. On the other hand, in the case of oncological patients, it is often necessary to use broad-spectrum antibiotics due to the reduced ability of the immune system to fight infections and the resulting co-occurrence of bacterial infections of various etiologies. For this reason, the possibility of using agents targeting a specific species of pathogenic bacteria is a promising perspective for the future because such treatment would spare the physiological intestinal microflora [47].

The symptoms of IFI are usually nonspecific. This is particularly important in cancer patients because, due to the reduced immunity, IFI may develop with only a few symptoms: refractory fever and increased markers of an ongoing infection (CRP and PCT), which do not respond to broad-spectrum antibiotic therapy [44,48,49,50,51]. In our case, the patient had persistent fever, abnormal CRP and PCT values, and neutropenia; therefore, a decision was made to extend the diagnostics towards IFI. A diagnostic challenge in our case was a positive blood test result for *C. albicans* and a CT scan depicting abnormalities in the lungs. The result of the imaging examination in our patient did not correlate with a typical clinical picture of *C. albicans* infection. Most often, yeast-like fungal infections manifest themselves as beige-white deposits occupying mainly the oral cavity and esophagus, and their presence causes discomfort (burning or pain) or a cough in patients [52,53]. Moreover, patients with malignant diseases of the hematopoietic system, due to chronic neutropenia, may develop chronic disseminated candidiasis (CDC), which often affects the liver and the spleen (hepatosplenic candidosis (HSC) [54,55,56]. Graeter et al. report the case of an 18-year-old patient with relapsed B-cell ALL who developed multi-organ CDC during re-induction chemotherapy complicated by neutropenic fever. Based on the skin punch biopsy results, a diagnosis of *Candida tropicalis* was made. After antifungal treatment, clinical improvement was achieved; however, a pre-transplant CT scan revealed multiple hepatosplenic and right kidney lesions and a nodular-appearing opacity in the lateral left lower lobe. Although the lung nodule biopsy turned out to be negative for fungal growth, the biopsy result of the liver lesions confirmed an infection with *Candida* species [57]. It is worth emphasizing that even in the course of disseminated candidiasis, cases of lung infections are rare [58]. Furthermore, pulmonary candidiasis itself is difficult to diagnose. The results of imaging studies in the case of lungs being affected by *Candida* spp. are nonspecific, presenting mainly bilateral nodules, often with an area of airspace consolidation [58,59]. In turn, the isolation of yeast from respiratory tract secretions, sputum or bronchoalveolar lavage fluid, may indicate the colonization or contamination of the tube. Therefore, the final diagnosis of pulmonary candidiasis is dependent on the result of a histopathological examination of a lung tissue sample taken during a biopsy [60].

The use of an invasive diagnostic procedure in the described case was crucial because the result of the histopathological examination of the lung biopsy taken from the patient allowed us to confirm an infection caused by *Aspergillus* spp. In contrast to *Candida* fungi, *Aspergillus* infections are most often located in the upper respiratory tract, bronchi and lung parenchyma [61,62,63]. Therefore, patients with *Aspergillus* spp. infection most often report respiratory symptoms: mild hemoptysis, chest pain, dyspnea and rapid breathing [64]. It is worth emphasizing that factors secreted by *Aspergillus* induce platelet activation, which may lead to hemorrhagic and necrotic changes in the lung parenchyma [65]. 

The diagnosis of suspected pulmonary aspergillosis in children is based on imaging tests [66,67]. The European Organization for Research and Treatment of Cancer/Invasive Fungal Infections Cooperative Group (EORTC/MSG) has developed criteria for confirming the diagnosis of pulmonary aspergillosis based on CT scans [68]. In children with hematological malignancies, the most frequently described ones are as follows: nodular lesions (59–100%) and wedge, segmental, or lobar consolidations (21–63%). Atypical lesions, such as pleural effusion and ground-glass opacities, are observed in only 2.5% of patients [69]. However, in our case, the CT scan was equivocal, showing, among other things, ground-glass opacities. For this reason, it was decided to perform a lung biopsy. A similar case of a 15-year-old girl with acute myeloid leukemia, who had confirmed *Aspergillus* infection, was described by Aljutaily et al. The initial images were nonspecific. The patient had a cough, fever and febrile neutropenia. Due to respiratory distress, radiological imaging and bronchoscopy were performed. A chest X-ray showed lower lobe opacities, but bronchoalveolar lavage did not reveal the source of the infection. Due to the patient’s deteriorating condition and their lack of response to the treatment, a thoracoscopy was performed. Histopathology confirmed *Aspergillus fumigatus* etiology [70]. It is worth emphasizing that some lesions can be detected much earlier with the use of positron emission tomography (PET) with 18F-fluorodeoxyglucose [71]. This is an imaging method that allows the detection of fungal lesions due to the increased glucose metabolism in tissues affected by inflammation [72,73].

Early therapeutic intervention in the case of fungal infections is crucial to achieve the desired clinical effect and prevent the development of complications [74,75]. Initially, empirical treatment is used [45]. This treatment regimen is mainly applied in children with neutropenia and persistent fever in >96 h, despite the administration of broad-spectrum antibiotics [76,77]. Empirical antifungal therapy may also be considered in the case of patients with persistent fever, low-risk disease, or profound and persistent granulocytopenia [14]. The drug used in this type of therapy (regardless of the age of the patient) is caspofungin or liposomal amphotericin B [78,79,80]. There are currently no guidelines specifying the duration of empirical therapy. Some studies indicate that drugs should be administered until the patient’s condition improves and the neutrophil level is equalized [14]. PET can also be used to assess the response to treatment and thus determine the moment at which it can be safely discontinued [81,82]. On the one hand, empirical antifungal treatment reduces the risk of IFI, and on the other hand, it may lead to drug resistance as well as increasing treatment costs [77,83,84]. Additionally, the prolonged administration of antifungal treatment introduces the possibility of a spectrum of adverse events that may not occur in the case of a shorter therapy course [85]. In patients at risk of developing IFI, in addition to antifungal therapy, probiotic supplementation may be considered in order to reduce the risk of developing IFI [86]. Studies have reported benefits of using probiotics containing symbiotic bacteria that can disrupt mycelium growth and inhibit the adhesion of pathogenic fungi and biofilm formation [87].

In cases of confirmed candidiasis, targeted treatment is implemented, taking into account the severity of the infection, the effectiveness of antifungal treatment in previous episodes, the intolerance of antifungal drugs and the drug resistance of a given fungal species [88]. For pediatric patients, the recommended first-line treatment is echinocandins or liposomal amphotericin B [89]. Both drugs have a high safety profile and can be used in the treatment of virtually all patients [90,91,92]. In the described case, the patient received liposomal amphotericin B and a drug from the echinocandins group—caspofungin. Thanks to the use of the aforementioned therapeutic agents, the patient achieved clinical improvement. In the absence of improvement, the second-line drugs are voriconazole and fluconazole [14]. The administration of glucocorticosteroids may be an alternative for patients with disseminated candidiasis who do not respond to antifungal therapy. Due to the immune system reaction induced by Candida infection, which causes the development of systemic inflammation, the use of glucocorticosteroids may result in a faster resolution of symptoms and reduction in inflammation indicators [93,94,95,96,97,98]. Additionally, in the case of disseminated hepatosplenic candidiasis, azoles are used in the treatment with good clinical response, mainly fluconazole or voriconazole [99]. Amphotericin B is also a good choice due to its potential accumulation in the reticuloendothelial system [100]. The duration of treatment depends on the severity of the fungal disease, the patient’s general condition and their response to treatment. In the invasive treatment of candidiasis, it is assumed that the therapy should last up to 14 days; however, the Infectious Diseases Society of America (IDSA) guidelines state that the therapy of hepatosplenic candidiasis should be continued until the lesions disappear on repeat imaging, which usually occurs after several months. The untimely discontinuation of the therapy is associated with a high risk of relapses [14,101,102,103]. The optimization of the antifungal therapy is extremely important because it may prevent the need for modifications to the therapeutic protocol. In particular, treatment with CDC should not delay a hematopoietic stem cell transplantation [104].

In turn, the first-line drug in the treatment of invasive aspergillosis in children over two years of age is voriconazole administered intravenously. In younger patients, the recommended treatment is liposomal amphotericin B with therapeutic drug concentration monitoring [14,105,106]. Alternative therapies include posaconazole, isavuconazole, and echinocandins [107]. In the case of isolated aspergillosis limited to organs, surgical treatment may be necessary [108]. There are no clear guidelines for the duration of aspergillosis treatment; however, the IDSA recommend that the therapy be continued for at least 6–12 weeks, taking into account the patient’s clinical condition and response to the applied therapy [14,101,102,109].

In recent years, new antifungal drugs and therapies have emerged, primarily with a view to addressing the problem of resistance to treatment [110,111,112]. Agents with different mechanisms of action are being developed, e.g., inhibiting biofilm formation [113,114]. Research is also being carried out on immunotherapy, which involves stimulating the immune system to target and eliminate fungal infections [115]. These include the use of antifungal vaccines, immunomodulatory drugs, and monoclonal antibodies [116,117,118,119,120]. Immunotherapy implemented in the early phase of treatment may reduce the need for intensive chemotherapy, and the use of innovative immunochemotherapy regimens may lead to hematological as well as molecular remission in patients with ALL [121]. However, more research is needed to determine the safety and effectiveness of these therapies in children.

## 5. Conclusions

The early application of diagnostic procedures may be helpful to identify the origin of the infection. Co-infections should be taken into account in the case of prolonged infection episodes. Whenever there is no improvement in the patient’s general condition and remission is not observed in radiological imaging findings, it may be beneficial to consider performing invasive diagnostic procedures. The prompt initiation and personalization of the targeted antifungal therapy is crucial to overcome IFI. In order to increase the effectiveness of treatment and reduce adverse events, it is important to adapt the therapeutic method to the clinical and metabolic characteristics of the patient. It is also important to use a personalized approach to effectively prevent the development of infection. In the future, this can be achieved, among other things, thanks to knowledge of the composition of the patient’s physiological microflora and mycoflora and their appropriate modulation.

## Figures and Tables

**Figure 1 pathogens-13-00772-f001:**
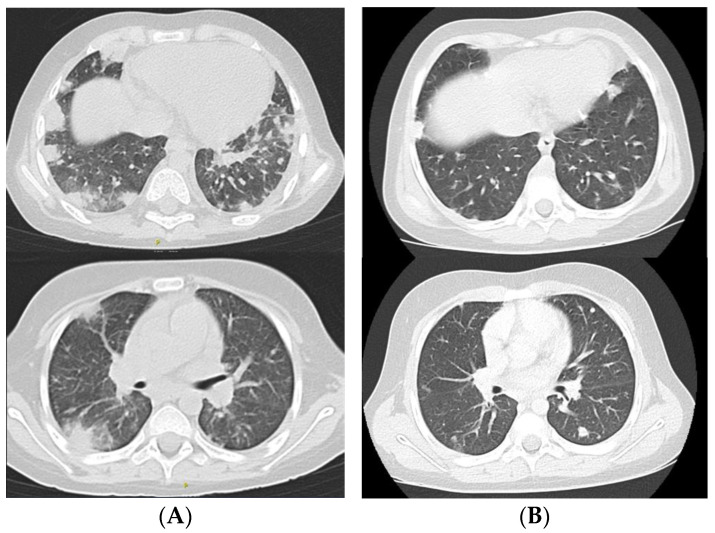
Axial sections showing multiple nodular and small nodular densities with the halo sign in both lungs. Features are consistent with fungal etiology. (**A**) The computed tomography findings—an examination conducted at the onset of the respiratory symptoms (first episode); (**B**) the computed tomography findings—a regression of pulmonary lesions following the implementation of effective antifungal therapy (second episode).

**Figure 2 pathogens-13-00772-f002:**
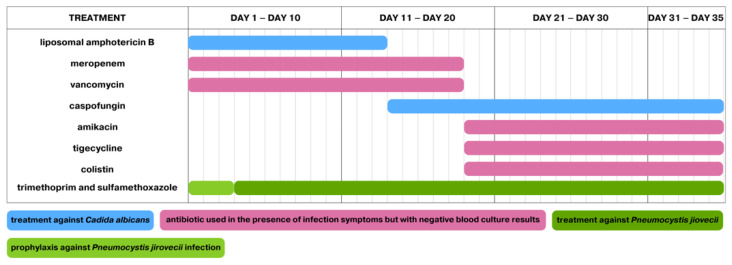
Treatment during the first episode of infection. The first day of treatment is also the first day of infection.

**Figure 3 pathogens-13-00772-f003:**
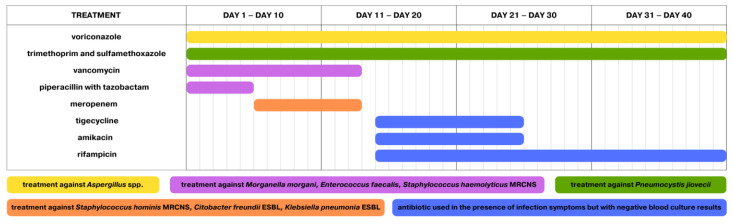
Treatment during the second episode of infection. The first day of treatment is also the first day of infection.

**Figure 4 pathogens-13-00772-f004:**
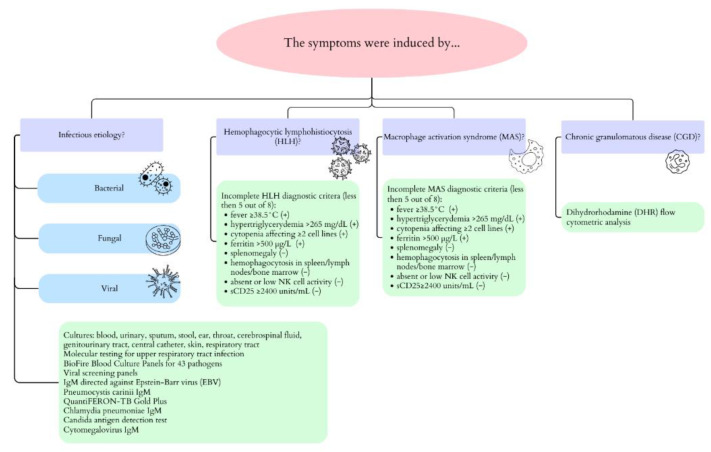
Diseases considered in differential diagnosis and the conducted tests.

## Data Availability

The data are available from the corresponding author upon reasonable request. Due to privacy reasons, the data are not publicly available.

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
