# Peer review of "Diagnostic and Therapeutic Challenge Caused by Candida albicans and Aspergillus spp. Infections in a Pediatric Patient as a Complication of Acute Lymphoblastic Leukemia Treatment: A Case Report and Literature Review"

_pathogens, 2024, doi:10.3390/pathogens13090772_

Round 1

Reviewer 1 Report (Previous Reviewer 1)

Comments and Suggestions for Authors

Discussion is too long (174 lines). You should reduce it indicating substantial  information related to major results. 

Author Response

Reviewer 2 Report (Previous Reviewer 2)

Comments and Suggestions for Authors

The authors have made the requested corrections and rectifications. A few comments may further enrich the article.

      Lines 252-256 : the patient described having received steroids and chemotherapeutics causing immunosuppression, then  treatment based on taking antibiotics. It seems difficult under these conditions to know whether a reduction in the number of commensal bacteria and fungi is due to oncological treatment or antibiotic therapy or both at the same time? The sentence between lines 252-256 should be reworded along these lines.     Line 347:  Can you cite publications that discuss the benefit of discontinuous antifungal prescriptions? The use of pre or probiotics can bring improvements, particularly preventatively.

 Line 404 : Knowledge of the composition of the microbiota of patients before treatments is undoubtedly one of the ways to follow in order to prevent infections (bacterial, viral and fungal inseparable). This is part of the rapid customization of the antifungal treatment. preventive treatment remains the best approach, in order to increase the effect

Author Response

This manuscript is a resubmission of an earlier submission. The following is a list of the peer review reports and author responses from that submission.

Round 1

Reviewer 1 Report

Comments and Suggestions for Authors

1. How frequent are infections, particularly invasive, by bacteria, fungi or virus in the hospital were the pediatric patient was attended?. I ask this because you mention % data for other hospitals. 

2. Right part of Fig. 1 is 1B not 2B.

3. Figs. 2 and 3: in the column referring to days of treatment, you should let clear that they are days post-infection.

4. Discussion is 387 lines long. Of these, about half are dedicated to discuss  aspects such as epidemiology, risk factors, treatment, etc. of the organisms you detected. Rigurously, this is not a discussion but rather a review of lierature on this organisms.  

Reviewer 2 Report

Comments and Suggestions for Authors

Review

Journal Pathogens (ISSN 2076-0817)

Manuscript ID pathogens-3008572

Type Review

Title Diagnostic and therapeutic challenge caused by Candida albicans and Aspergillus spp. infections in pediatric patient as a complication of acute lymphoblastic leukemia treatment: a case report and literature review

Authors Natalia Zaj , Weronika Kopyt , Emilia Kamizela , Monika Lejman , Joanna Zawitkowska *

Section Fungal Pathogens

   This article presents the case of a 5-year-old boy who suffered from ALL of B-cell precursor origin. An infectious complication caused by Candida albicans (C. albicans) and Aspergillus spp occurred during the consolidation phase of ALL treatment. . The aim of this article is to address the diagnostic and therapeutic challenges of fungal infections. The clinical presentation of the case and its treatment is also proposed.

   Chronic disseminated candidiasis is an invasive fungal infection observed during neutrophil recovery in patients with acute leukemia treated with intensive chemotherapy.

 Already several publications suggest that in children with persistent symptomatic ( CDC), with adequate antifungal treatment, the administration of corticosteroids may result in rapid resolution of symptoms. However, in patients who do not respond to steroids, the addition of a nonsteroidal anti-inflammatory drug is considered effective. Shkalim-Zemer V. Pediatr Infect Dis J. 2018.or more recent  Colin-Benoit E, Infection. 2023 .

 Currently the treatment of ALL remains empirical because many unknowns remain There are different gene expression in ALL caused by a broad range of genetic alterations. Ideally individual identification of mutations, depending on drugs susceptibility testing represents a panel of chemotherapeutic agents. This research represents hope to identify individual therapeutic vulnerabilities and appropriate drug combinations. Ideally the search for a reduced-intensity conventional immunotherapy  or a regimen without chemotherapy, which may ultimately improve patient survival while reducing harmful effects, constitutes the objective of the current research. Inaba H,  Haematologica. 2020 Nov 1;105(11):2524-2539.

  In this context, the proposed article lacks on the references  details of the protocols implemented. There are many therapeutic possibilities in the presence of different  ALL in pediatrics.

Detailed remarks.

 Line 21 : what reference for protocol II?   Line 78 : All these references must be linked to referenced therapeutic protocols.   Line 79 : ampiric therapy is based on what reference ?   Line 84 : Can you provide more details about these products?

Line 89 : other possible  treatment with clindamycin and primaquine due to history of significant allergy to sulfa drugs. ref à préciser Springsted E, Giri B, Kollipara V. Pneumocystis jirovecii pneumonia in newly diagnosed treatment-naïve chronic lymphocytic leukaemia. BMJ Case Rep. 2021 Jun 23;14(6):e241888. doi: 10.1136/bcr-2021-241888. PMID: 34162609; PMCID: PMC8230998.

Line 152 : why this choice "This time, tigrecycline, amikacin and ri-fampicin were implemented".

Line 245 : the possibility of targeting pathogenic bacteria still remains hypothetical, yet this is undoubtedly what we should aim for.

 Line 252 : this remark deserves to be developed with references.

Line 379 :  Does this paragraph relate to the case presented in the article or is it corroborated by other publications ?

 Line 394 : In the conclusion, we must mention the need to adapt therapy on a case-by-case basis. Currently it seems difficult to offer a precise treatment that suits all cases.

Additional library

Lussana F, Cavallaro G, De Simone P, Rambaldi A. Optimal Use of Novel Immunotherapeutics in B-Cell Precursor ALL. Cancers (Basel). 2023 Feb 20;15(4):1349. doi: 10.3390/cancers15041349. PMID: 36831690; PMCID: PMC9954469.

Shkalim-Zemer V, Levi I, Fischer S, Tamary H, Yakobovich J, Avrahami G, Gilad G, Elitzur S, Yaniv I, Elhasid R, Manistersky M, Shalit I. Response of Symptomatic Persistent Chronic Disseminated Candidiasis to Corticosteroid Therapy in Immunosuppressed Pediatric Patients: Case Study and Review of the Literature. Pediatr Infect Dis J. 2018 Jul;37(7):686-690. doi: 10.1097/INF.0000000000001844. PMID: 29140934.

Inaba H, Mullighan CG. Pediatric acute lymphoblastic leukemia. Haematologica. 2020 Nov 1;105(11):2524-2539. doi: 10.3324/haematol.2020.247031. PMID: 33054110; PMCID: PMC7604619.

Colin-Benoit E, Kalubi M, Zimmerli S. A case of chronic disseminated candidiasis in metamizole-induced neutropaenia. Infection. 2023 Jun;51(3):775-778. doi: 10.1007/s15010-022-01963-z. Epub 2022 Dec 14. PMID: 36515891; PMCID: PMC9748381.

Rammaert B, Maunoury C, Rabeony T, Correas JM, Elie C, Alfandari S, Berger P, Rubio MT, Braun T, Bakouboula P, Candon S, Montravers F, Lortholary O. Does 18F-FDG PET/CT add value to conventional imaging in clinical assessment of chronic disseminated candidiasis? Front Med (Lausanne). 2022 Dec 20;9:1026067. doi: 10.3389/fmed.2022.1026067. PMID: 36606049; PMCID: PMC9807873.

As part of a review of the literature on this subject, the references to follow are lacking but nevertheless have their place.

Rammaert B, Desjardins A, Lortholary O. New insights into hepatosplenic candidosis, a manifestation of chronic disseminated candidosis. Mycoses. (2012) 55:e74–84. 10.1111/j.1439-0507.2012.02182.x [PubMed] [CrossRef] [Google Scholar]

 Chen C-Y, Cheng A, Tien F-M, Lee PC, Tien HF, Sheng WH, et al. Chronic disseminated candidiasis manifesting as hepatosplenic abscesses among patients with hematological malignancies. BMC Infect Dis. (2019) 19:635. 10.1186/s12879-019-4260-4 [PMC free article] [PubMed] [CrossRef] [Google Scholar]

 Grateau A, Le Maréchal M, Labussière-Wallet H, Ducastelle-Leprêtre S, Nicolini FE, Thomas X, et al. Chronic disseminated candidiasis and acute leukemia: impact on survival and hematopoietic stem cell transplantation agenda. Med Mal Infect. (2018) 48:202–6. 10.1016/j.medmal.2017.12.004 [PubMed] [CrossRef] [Google Scholar]

Pappas PG, Kauffman CA, Andes DR, Clancy CJ, Marr KA, Ostrosky-Zeichner L, et al. Clinical practice guideline for the management of candidiasis: 2016 update by the infectious diseases society of America. Clin Infect Dis. (2016) 62:e1–50. [PMC free article] [PubMed] [Google Scholar]

 Ullmann AJ, Akova M, Herbrecht R, Viscoli C, Arendrup MC, Arikan-Akdagli S, et al. ESCMID* guideline for the diagnosis and management of Candida diseases 2012: adults with haematological malignancies and after haematopoietic stem cell transplantation (HCT). Clin Microbiol Infect. (2012) 18(Suppl 7):53–67. 10.1111/1469-0691.12041 [PubMed] [CrossRef] [Google Scholar]

 Jang Y-R, Kim M-C, Kim T, Chong YP, Lee SO, Choi SH, et al. Clinical characteristics and outcomes of patients with chronic disseminated candidiasis who need adjuvant corticosteroid therapy. Med Mycol. (2018) 56:782–6. 10.1093/mmy/myx110 [PubMed] [CrossRef] [Google Scholar]

 De Castro N, Mazoyer E, Porcher R, Raffoux E, Suarez F, Ribaud P, et al. Hepatosplenic candidiasis in the era of new antifungal drugs: a study in Paris 2000-2007. Clin Microbiol Infect. (2012) 18:E185–7. 10.1111/j.1469-0691.2012.03819.x [PubMed] [CrossRef] [Google Scholar]

 Legrand F, Lecuit M, Dupont B, Bellaton E, Huerre M, Rohrlich PS, et al. Adjuvant corticosteroid therapy for chronic disseminated candidiasis. Clin Infect Dis. (2008) 46:696–702. [PubMed] [Google Scholar]

Dellière S, Guery R, Candon S, Rammaert B, Aguilar C, Lanternier F, et al. Understanding pathogenesis and care challenges of immune reconstitution inflammatory syndrome in fungal infections. J Fungi. (2018) 4:E139. [PMC free article] [PubMed] [Google Scholar]

 Candon S, Rammaert B, Foray AP, Moreira B, Gallego Hernanz MP, Chatenoud L, et al. Chronic disseminated candidiasis during hematological malignancies: an immune reconstitution inflammatory syndrome with expansion of pathogen-specific T helper type 1 cells. J Infect Dis. (2020) 221:1907–16. 10.1093/infdis/jiz688 [PubMed] [CrossRef] [Google Scholar]

 Koh KC, Slavin MA, Thursky KA, Lau E, Hicks RJ, Drummond E, et al. Impact of fluorine-18 fluorodeoxyglucose positron emission tomography on diagnosis and antimicrobial utilization in patients with high-risk febrile neutropenia. Leuk Lymphoma. (2012) 53:1889–95. 10.3109/10428194.2012.677533 [PubMed] [CrossRef] [Google Scholar]

 Chamilos G, Macapinlac HA, Kontoyiannis DP. The use of 18F-fluorodeoxyglucose positron emission tomography for the diagnosis and management of invasive mould infections. Med Mycol. (2008) 46:23–9. [PubMed] [Google Scholar]

 Mohan M, Fogel B, Eluvathingal T, Schinke C, Kothari A. Gastrointestinal histoplasmosis in a patient after autologous stem cell transplant for multiple myeloma. Transpl Infect Dis. (2016) 18:939–41. 10.1111/tid.12619 [PubMed] [CrossRef] [Google Scholar]

 Reyes N, Onadeko OO, Luraschi-Monjagatta MDC, Knox KS, Rennels MA, Walsh TK, et al. Positron emission tomography in the evaluation of pulmonary nodules among patients living in a coccidioidal endemic region. Lung. (2014) 192:589–93. 10.1007/s00408-014-9589-2 [PubMed] [CrossRef] [Google Scholar]
